

# Outlier detection for keystroke biometric user authentication

Mahmoud G. Ismail[1], Mohammed A.-M. Salem[1], Mohamed A. Abd El Ghany[2,3], Eman Abdullah Aldakheel[4] and Safia Abbas[5]

[1] Faculty of Media Engineering and Technology, German University in Cairo, Cairo, Egypt
[2] Electronics Department, German University in Cairo, Cairo, Egypt
[3] Integrated Electronic Systems Lab, Technische Universität Darmstadt, Darmstadt, Germany
[4] Department of Computer Sciences, College of Computer and Information Sciences, Princess Nourah Bint Abdulrahman University, Riyadh, Saudi Arabia
[5] Department of Computer Sciences, Faculty of Computer and Information Sciences, Ain Shams University, Cairo, Egypt

Corresponding author
Mahmoud G. Ismail,
mahmodg.ismail@gmail.com

## ABSTRACT

User authentication is a fundamental aspect of information security, requiring robust measures against identity fraud and data breaches. In the domain of keystroke dynamics research, a significant challenge lies in the reliance on imposter datasets, particularly evident in real-world scenarios where obtaining authentic imposter data is exceedingly difficult. This article presents a novel approach to keystroke dynamics-based authentication, utilizing unsupervised outlier detection techniques, notably exemplified by the histogram-based outlier score (HBOS), eliminating the necessity for imposter samples. A comprehensive evaluation, comparing HBOS with 15 alternative outlier detection methods, highlights its superior performance. This departure from traditional dependence on imposter datasets signifies a substantial advancement in keystroke dynamics research. Key innovations include the introduction of an alternative outlier detection paradigm with HBOS, increased practical applicability by reducing reliance on extensive imposter data, resolution of real-world challenges in simulating fraudulent keystrokes, and addressing critical gaps in existing authentication methodologies. Rigorous testing on Carnegie Mellon University's (CMU) keystroke biometrics dataset validates the effectiveness of the proposed approach, yielding an impressive equal error rate (EER) of 5.97%, a notable area under the ROC curve of 97.79%, and a robust accuracy (ACC) of 89.23%. This article represents a significant advancement in keystroke dynamics-based authentication, offering a reliable and efficient solution characterized by substantial improvements in accuracy and practical applicability.

Subjects Algorithms and Analysis of Algorithms, Artificial Intelligence, Data Mining and Machine Learning, Data Science, Security and Privacy
Keywords Keystroke biometrics, Machine learning, Outlier detection, User authentication, Histogram-based outlier score, Carnegie Mellon University's (CMU) keystroke biometric dataset

## INTRODUCTION

User authentication stands as a cornerstone in safeguarding information security, offering critical defense against unauthorized access to protected systems. In today's evolving landscape marked by identity fraud, data breaches, and the increasing adoption of remote work, the necessity for secure and validated remote access becomes ever more pronounced.

Within the realm of user authentication, three conventional modes emerge: possession, knowledge, and biometrics.

Possession authentication relies on tangible items such as keys, passports, or smart cards, historically serving as robust authentication mechanisms. However, these items are susceptible to vulnerabilities like sharing, duplication, loss, or theft. Knowledge authentication relies on confidential information, typically manifesting as passwords, yet faces issues such as predictability and susceptibility to sharing. Biometrics, encompassing physiological and behavioral traits, plays a pivotal role in security. It spans modalities such as facial recognition, fingerprint analysis, iris scans, and keystroke dynamics.

Keystroke dynamics, a subset of behavioral biometric authentication, focuses on the unique timing patterns generated during individuals' interactions with computer keyboards. It leverages the distinct rhythm and cadence exhibited during key presses and releases, creating a personalized signature of typing behavior. Originating from Morse code operators, where individuals displayed unique typing rhythms, keystroke dynamics has evolved into a sophisticated authentication method capturing individual differences in anatomy, typing habits, emotions, and context.

## Motivation

Keystroke dynamics offer a promising avenue for user authentication due to their inherent individuality, providing an additional layer of security beyond traditional password-based methods. However, the efficacy of keystroke biometrics relies heavily on the quality and authenticity of the data used for training and testing these systems.

This research is motivated by the critical need to overcome the significant challenges associated with obtaining authentic imposter data for keystroke biometric research. Ethical concerns and privacy limitations pose formidable obstacles to collecting real, diverse imposter data, as highlighted in recent studies (*Sadikan, Ramli & Fudzee, 2019*; *Monrose & Rubin, 2000*). The reliance on simulated fraudulent keystrokes in existing datasets often fails to accurately capture genuine imposter behavior, leading to unreliable outcomes.

Moreover, the inherent variability in typing patterns (*González et al., 2022*) and the potential for social engineering attacks underscore the necessity to enhance the robustness and generalizability of current keystroke biometric systems. Instead of addressing these challenges individually, it is more efficient to find solutions that do not require imposter data for training.

This research aims to develop an alternative approach, bypassing the need for imposter data and improving the resilience of keystroke biometrics against typing variability and social engineering threats. Through these advancements, we aim to unlock the full potential of keystroke dynamics as a secure and user-friendly authentication tool in real-world security applications.

## Contributions of the work

The main contributions of this work are:

1. **Alternative approach to keystroke authentication:** Introducing unsupervised outlier detection techniques, notably histogram-based outlier score (HBOS), as an alternative

to traditional imposter dataset reliance, establishing a new paradigm in keystroke dynamics research.

2. **Enhanced practical applicability:** Improving practical applicability by bypassing the need for extensive imposter datasets, addressing scenarios where data collection is impractical or privacy concerns prevail.

3. **Addressing real-world challenges and methodological gaps:** Tackling challenges in simulating fraudulent keystrokes and limitations of existing approaches, filling a crucial gap in authentication methodologies with a more robust solution aligned with real-world constraints.

## RELATED WORK

Keystroke dynamics research has witnessed a surge in the application of diverse machine learning methodologies, including nearest neighbor classifiers, K-means, Bayesian classifiers, bagging, and boosting. Noteworthy studies include the utilization of the K-nearest neighbor (KNN) algorithm with dependence clustering, achieving a 7.7% equal error rate (EER) (*Ivannikova, David & Hämäläinen, 2017*). The generalized fuzzy model (GFM), incorporating Gaussian mixture models, demonstrated a 7.86% EER (*Bhatia et al., 2018*). A modified differential evolution (MDE)-based subspace anomaly detection mechanism exhibited an impressive 3.48% EER (*Krishna & Ravi, 2019*). In another study, the utilization of the KNN algorithm with dimensionality reduction achieved an accuracy of 87.5% (*Sahu, Banavar & Schuckers, 2020*). Support vector machines, random forests, and neural networks were explored, with neural networks emerging as the top performer, boasting an accuracy of 91.8% (*Thakare et al., 2021*). Additionally, a novel barcoding system employing one-class support vector machines (SVM) yielded promising outcomes, achieving a minimum EER of 9.88% (*Alpar, 2021*). The application of X-means clustering resulted in an impressive area under the ROC curve (AUC) of 94.2% and an EER of 11.2% (*Hazan, Margalit & Rokach, 2021*). Extreme Gradient Boosting (XGBoost) and multi-layer perceptrons (MLP) showcased robust performance, with an augmented XGBoost model achieving a noteworthy accuracy of 96.39% (*Chang et al., 2022*). A hybrid POHMM/SVM technique, averaging an EER of 8.6%, was also reported in *Ali & Tappert (2018)*. Notably, the utilization of quantile transformation for normalizing keystroke timing, combined with histogram gradient boosting, delivered compelling results, boasting an impressive 97.96% accuracy and a 1.4% EER (*Ibrahim et al., 2023*).

In the realm of deep neural networks, strategies such as Adam optimization and LeakyReLU activation, as highlighted by *Maheshwary, Ganguly & Pudi (2017)*, are employed to expedite learning. This typically involves three hidden layers with 100, 400, and 100 units, LeakyReLU in the hidden layers, and Softmax activation in the output layer, resulting in an impressive 3% EER and 93.59% accuracy. Independent learning strategies, incorporating multiple layers and the Nadam optimizer, achieve a notable accuracy of 92.60%, as demonstrated by *Muliono, Ham & Darmawan (2018)*. For keystroke authentication enhancement, *Patel et al. (2019)* employs an autoencoder with two phases, yielding a 6.51% EER. Another approach presented by *Andrean, Jayabalan*

**Table 1  Recent works using Carnegie Mellon University's (CMU) keystroke benchmark dataset sorted by publication year.**

| Author | Approach | EER | Accuracy |
|---|---|---|---|
| *Ibrahim et al. (2023)* | Histogram Gradient Boosting | 1.4% | 97.96% |
| *Chang et al. (2022)* | XGBoost-augment | _ | 96.39% |
| *Ali & Tappert (2018)* | POHMM+SVM | 8.6% | _ |
| *Thakare et al. (2021)* | ANN | _ | 91.8% |
| *Alpar (2021)* | Scalogram Barcoding and One-class SVM | 9.88% | _ |
| *Hazan, Margalit & Rokach (2021)* | X-means with QT | 11.2% | _ |
| *Sahu, Banavar & Schuckers (2020)* | Kernel PCA with KNN | _ | 87.5% |
| *Andrean, Jayabalan & Thiruchelvam (2020)* | MLP | 4.45% | _ |
| *Gedikli & Efe (2020)* | Feed Forward Neural Network | _ | 94.7% |
| *Patel et al. (2019)* | Autoencoder model | 6.51% | _ |
| *Krishna & Ravi (2019)* | Modified Differential Evolution | 34.8% | _ |
| *Muliono, Ham & Darmawan (2018)* | Deep Learning Nadam optimizer | _ | 92.60% |
| *Bhatia et al. (2018)* | Generalized Fuzzy Model (GFM) | 7.86% | _ |
| *Ivannikova, David & Hämäläinen (2017)* | Dependence Clustering + KNN | 7.7% | _ |
| *Maheshwary, Ganguly & Pudi (2017)* | Deep Secure | 3% | 93.59% |

*& Thiruchelvam (2020)* utilizes a MLP with an input layer (31 neurons) and two hidden layers (23 neurons), achieving an EER of 4.45%. Additionally, in a study by *Gedikli & Efe (2020)*, three feed-forward neural network models were explored, with the first model's configuration (three hidden layers: 20, 30, 20 neurons) and Rprop mechanism achieving a 4.9% EER and an average identification accuracy of 94.7%.

To summarize the findings of previous works, Table 1 presents recent advances using Carnegie Mellon University's (CMU) keystroke dataset, sorted by publication date.

The reliance on imposter datasets for training poses a significant limitation in keystroke dynamics research, especially in real-world scenarios where obtaining authentic imposter data is challenging and may not accurately represent actual imposter behavior. Existing approaches often rely on simulated fraudulent keystrokes for evaluation, highlighting a gap in practical applicability (see Table 1). In response, this research addresses this challenge by exploring unsupervised outlier detection techniques, such as the HBOS. This alternative approach bypasses the need for extensive imposter datasets, offering a promising solution to enhance the practical application of keystroke dynamics-based authentication. The goal is to provide a more robust and applicable solution for real-world security scenarios, overcoming the limitations associated with imposter data collection.

## METHODOLOGY

In this research, the primary focus was on exploring unsupervised approaches for keystroke dynamics authentication, primarily due to their effectiveness without the need for an imposter dataset, a common limitation in real-world applications. For instance, in scenarios where new employees receive unique usernames and passwords to access a company's database, the absence of a dataset containing multiple imposters with their distinctive

keystrokes recorded makes it challenging to employ supervised learning approaches. In contrast, unsupervised methods only necessitate genuine user data for effective training.

The approach rested on the premise that attackers or individuals attempting unauthorized access exhibit typing biometrics distinct from authorized users. Consequently, the typing patterns of potential attackers are considered outliers in comparison to the typical data patterns of legitimate users. Outliers represent extreme data points that significantly deviate from the expected norms within their respective categories. In the context of the research, data points marked as outliers effectively denoted potential attackers.

Utilizing the concept of outlier detection is a crucial process for identifying and managing data points that fall far from the data's average or are inconsistent with the norm. The removal or resolution of outliers is essential to prevent any potential skewing in the analysis. In this context, the identification of outliers corresponded to the detection of potential attackers within the keystroke dynamics dataset.

Expanding on the effectiveness demonstrated by histogram gradient boosting (HGB) in keystroke user authentication (*Ibrahim et al., 2023*), there is a compelling case for advocating the adoption of the HBOS. To assess the suitability of outlier detection techniques in the context of keystroke dynamics, a comprehensive study was conducted, evaluating the performance of various machine learning outlier detection models. Through rigorous testing involving 15 models, the HBOS emerged as the top-performing model.

The workflow of the keystroke user authentication system is depicted in Fig. 1. This process initiates with the registration of a password and involves user authentication based on their keystroke biometrics. The algorithm extracts distinctive features from the user's typing patterns, including Hold Time (H.Time) showing the duration a key is pressed, keyDown-keyDown time (DD.Time) which is the duration between two successive key presses, and keyUp-keyDown time (UD.Time) representing the period from key release to the press of another key. While additional features could potentially enhance authentication, their implementation often necessitates specialized hardware, such as measuring the force applied during a key press. This could significantly escalate the system's cost and limit its practicality for widespread use.

## Outlier detection models

Several outlier detection algorithms have been developed over the past decade, categorized into four groups: Linear outlier detection models, proximity-based outlier detection models, probabilistic outlier detection models, and ensembles and combination frameworks. This subsection details the 15 algorithms employed for user identification.

### Linear models for outlier detection

In the category of Linear models for outlier detection, three distinct methodologies were employed. Principal component analysis (PCA) utilizes a lower-dimensional space to generate principal components, assigning outlier scores based on distances from these components (*Shyu et al., 2003*). Mathematically, PCA involves the computation of principal components ($PC_i$) using the covariance matrix ($Cov$):

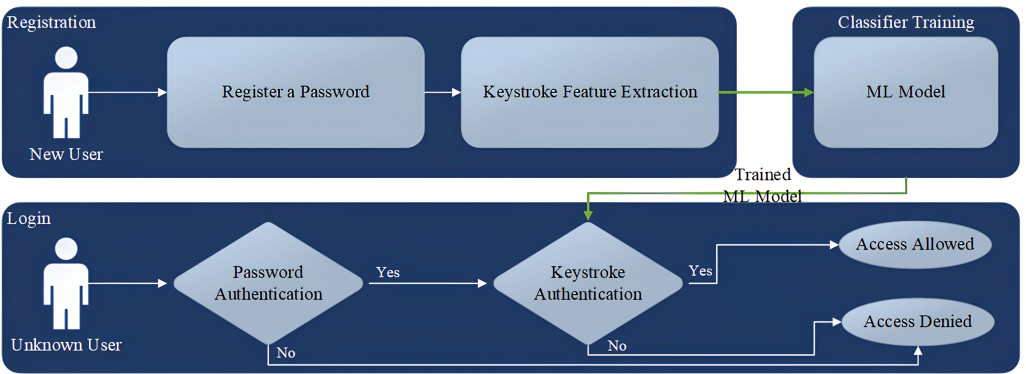

**Figure 1** **Keystroke user authentication flowchart.**

$PC_i = Cov(X)$.

One-Class support vector machines (OCSVM) map data into a high-dimensional space, constructing a hyperplane for maximal separation between normal instances and the origin, serving as a boundary to distinguish normal from abnormal instances (*Tax & Duin, 2004*). The optimization problem for OCSVM involves finding the hyperplane parameters ($w$ and $b$) that maximize the margin and minimize the classification error:

$\min \frac{1}{2}\|w\|^2 + \frac{1}{vn}\sum_{i=1}^{n}\xi_i - \rho$.

Linear model deviation-based outlier detection (LMDD) takes a unique approach, not relying on statistical analysis or distance-based metrics. It identifies outliers by assessing the primary properties of objects in a group, flagging those that 'deviate' from the group's description, with the term 'deviation' characterizing outliers (*Arning, Agrawal & Raghavan, 1996*). While the specific mathematical formulation for LMDD is not provided here, it involves deviation-based scoring for outlier identification.

### *Proximity-based outlier detection models*

Various approaches offer unique outlier detection methodologies within this category. Local Outlier Factor (LOF) assesses an object's local density deviation from its neighbors (*Breunig et al., 2000*). It utilizes K-nearest neighbors to estimate local density and identify outliers with significantly lower density, calculated as:

$LOF(p) = \frac{\sum_{o \in N_k(p)} \frac{lrd_k(o)}{lrd_k(p)}}{k}$

where $lrd_k(o)$ denotes the local reachability density of object $o$.

Optimizing computational costs, clustering-based local outlier factor (CBLOF) partitions data into clusters and applies LOF based on cluster size and distance to the nearest cluster (*He, Xu & Deng, 2003*).

KNN calculates the maximum distance of the kth neighbor as the outlier score (*Angiulli & Pizzuti, 2002*), given by:

Outlier Score$(p) = \max_{i=1}^{k} \text{distance}(p, \text{neighbor}_i)$

Mean KNN derives the score from the mean distance to the K-nearest neighbors as:

Outlier Score$(p) = \frac{\sum_{i=1}^{k} \text{distance}(p, \text{neighbor}_i)}{k}$

Assuming feature independence, the HBOS normalizes histograms, calculates the score for each feature, and flips values for outlier prioritization (*Goldstein & Dengel, 2012*), formulated as:

$$\text{HBOS}(p) = \prod_{i=1}^{d} \frac{1}{\text{hist}_i(b(p_i))}$$

where $\text{hist}_i$ is the normalized histogram for feature $i$ and $(p_i)$ is the bin index of feature $p_i$. Choosing the most appropriate method depends on specific needs, considering the strengths and weaknesses of each approach.

### Probabilistic models for outlier detection

This section explores three distinct probabilistic models employed for outlier detection: Angle-based outlier detection (ABOD), kernel density estimation (KDE), and Gaussian mixture model (GMM). Each model offers unique advantages and drawbacks, and their mathematical equations are presented below.

ABOD leverages the angle spectrum of data points to quantify their deviation from the majority (*Kriegel, Schubert & Zimek, 2008*), calculating the variance of weighted directional vectors. The angle between two points is given by $\angle(p_i, p_j) = \arccos(\frac{p_i \cdot p_j}{||p_i|| \cdot ||p_j||})$, where $p_i$ and $p_j$ are data points and $\cdot$ represents the dot product. The ABOD score for a point $i$ is calculated using the following formula:

$$\text{ABOD score for } p_i = \frac{\sum_{j \neq i} w_{ij}(\angle(p_i, p_j) - \overline{\angle(p_i)})^2}{\sum_{j \neq i} w_{ij}}.$$

In this equation, $w_{ij}$ represents the weight assigned to the angle between data points $p_i$ and $p_j$, and $\overline{\angle(p_i)}$ is the mean angle for point $p_i$.

KDE constructs a density profile by summing smooth kernels around each data point, estimating the probability density function (*Latecki, Lazarevic & Pokrajac, 2007*). The kernel density estimate at point $x$ is given by:

$$\hat{f}_h(x) = \frac{1}{nh} \sum_{i=1}^{n} K\left(\frac{x - x_i}{h}\right).$$

Here, $n$ is the number of data points, $h$ is the bandwidth parameter, $K$ is the kernel function, and $x_i$ represents the data points.

GMM assumes the data originates from a mixture of $k$ Gaussian distributions, using the expectation–maximization algorithm to estimate their parameters (*Aggarwal, 2016*). The mixture density model is given by:

$$p(x) = \sum_{k=1}^{K} \pi_k \mathcal{N}(x | \mu_k, \Sigma_k).$$

In this equation, $K$ is the number of Gaussian distributions, $\pi_k$ represents the mixing coefficients, $\mu_k$ is the mean vector, and $\Sigma_k$ is the covariance matrix for the $k$th Gaussian component.

### Outlier ensembles and combination frameworks

This section explores diverse strategies employed within outlier ensembles and combination frameworks to improve outlier detection performance.

Isolation forest: This method leverages an ensemble of isolation trees, identifying outliers as data points with shorter path lengths within the trees. While effective, it can struggle with local outliers due to multiple normal instance clusters, impacting isolation efficiency (*Liu, Ting & Zhou, 2008*).

Feature bagging: This approach consolidates outputs from multiple outlier detection algorithms, where each detector randomly selects a subset of features. An example includes an ensemble of 10 LOF classifiers with varying n_neighbors parameters (*Lazarevic & Kumar, 2005*).

Locally selective combination of parallel outlier ensembles (LSCP): This method defines local regions around data points based on nearest neighbors in randomly chosen feature sub-spaces. It then selects top-performing base detectors within these regions and combines them into the final model (*Zhao et al., 2019*).

Isolation-based anomaly detection using nearest-neighbor ensembles (INNE): This technique partitions the data space into regions using a subsample. It then assigns isolation scores based on local distributions for both global and local anomaly detection (*Bandaragoda et al., 2018*).

By understanding these diverse methods and their underlying mechanisms, researchers and practitioners can select the most suitable approach for their specific outlier detection needs, achieving more robust and accurate results.

## Hyper-parameter tuning

Hyper-parameter tuning is a crucial step in optimizing the performance of machine learning models. The goal of hyper-parameter tuning is to find the set of hyper-parameters that maximize the model's performance on a given dataset. Hyper-parameters are configuration settings that are external to the model and cannot be learned from the data. They control aspects such as the complexity of the model, the regularization strength, or the learning rate.

In our study, hyper-parameter tuning was conducted to enhance the effectiveness of various outlier detection models for keystroke biometric authentication. We selected a set of outlier detection models, including HBOS, isolation forest, PCA, LOF, and OCSVM, based on their suitability and widespread use in the field.

The tuning process involved systematically searching through a predefined hyper-parameter space to identify the combination that yields the best performance. For each model, we specified a range of values for its hyper-parameters, such as the number of bins for HBOS or the number of neighbors for LOF. We then used grid search technique to explore this parameter space efficiently.

During the tuning process, we evaluated the models using appropriate performance metrics, such as accuracy, precision, recall, F1-score, and area under the ROC curve (AUC). This evaluation helped us assess the models' performance under different hyper-parameter configurations and identify the optimal set of hyper-parameters that maximized performance.

By systematically tuning the hyper-parameters of the outlier detection models, we aimed to improve their accuracy in identifying outliers within keystroke biometric data. This

optimization process ensures that the models are well-suited for the specific task of user authentication, leading to more reliable and effective outlier detection.

## Carnegie Mellon University's keystroke biometric dataset

The evaluation of outlier detection approaches for user authentication utilized the CMU keystroke benchmark dataset (*Killourhy & Maxion, 2009*). This dataset, chosen for its comprehensive features and established performance metrics, contains keystroke-timing information from 51 subjects across eight sessions, with a one-day interval between sessions. Each session recorded keystroke data for 50 instances of typing the password '.tie5Roanl,' representing a strong password. The dataset encompasses three categories of features for each password character: hold time, down-down time, and up-down time.

Figure 2 provides an overview of the average timing data for three keystroke features across six randomly selected users from the CMU dataset. Figures 2A, 2B, and 2C represent average down-down timings, average up-down timings, and average hold keystroke timings, respectively.

Figure 2A illustrates significant variations in average down-down keystroke timings across different features and users, with noticeable inter-user variability within the same feature. Similarly, Fig. 2B showcases substantial diversity in up-down keystroke timings across features and users. In contrast, Fig. 2C explores average hold durations, revealing variations that are comparatively less pronounced than those observed in down-down and up-down timings.

These visualizations offer valuable insights into the intricate connection between individual writing habits and keystroke timing patterns, underscoring the pivotal role of individual writing habits in shaping keystroke dynamics. Furthermore, they highlight the potential significance of up-down keystroke sequences and hold duration as promising indicators for distinguishing user-specific characteristics within the selected user sample.

## RESULTS

### Experiment setting

The choice of input device significantly impacts keystroke dynamics and user authentication. Different devices introduce variability in typing behavior, affecting features extracted for authentication models. Familiarity with a device influences typing patterns and consistency. Users develop unique typing habits on familiar devices, but unfamiliar devices may lead to altered behavior, impacting model accuracy. Thus, to ensure consistency and comparability, the experiment focused on a controlled environment provided by the CMU dataset. This controlled setting limited participant variability to the same device (*i.e.,* keyboard) and the same password, enabling a more standardized evaluation. The experiment simulated a scenario where a long-term password had been compromised by an impostor. The primary objective was to measure how effectively each anomaly detector distinguished between the impostor's typing behavior and the genuine user's. The experiment followed these steps:

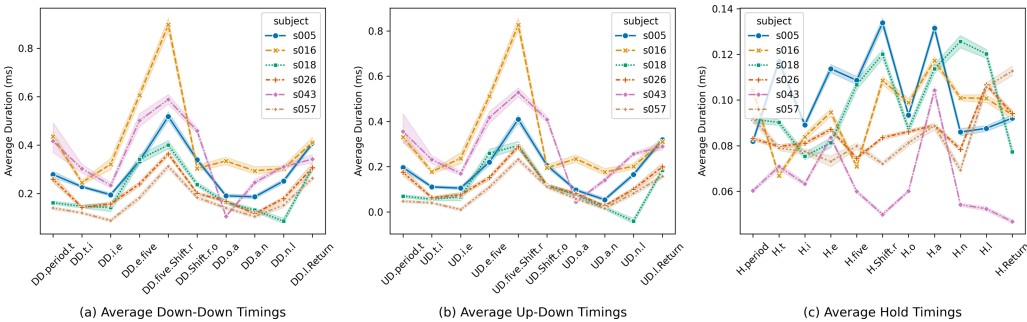

**Figure 2** Average keystroke timing data across selected users.

1. During the training phase, the detector analyzed the timing vectors from the first 200 password repetitions typed by the genuine user, constructing a model of the user's typing behavior.
2. In the test phase, the detector was run on the timing vectors from the subsequent 200 repetitions typed by the genuine user. Anomaly scores were recorded for each timing vector, constituting the user scores.
3. The test phase was repeated for the first five repetitions typed by each of the 50 impostors. Anomaly scores were recorded for their timing vectors, forming the impostor scores.

This process was iterated, designating each of the other subjects as the genuine user in turn. After training and testing each of the 15 detectors, a total of 765 sets of user and impostor scores were gathered (51 subjects × 15 detectors).

## Hardware specifications

The experiment was conducted on a computing platform equipped with the following hardware specifications:

- **Processor**: Intel (R) Core (TM) i7 -8750H CPU @ 2.20 GHz.
- **Installed RAM**: 16 GB.
- **Operating System**: Windows 10 64-bit.

## Evaluation metrics

While prior studies predominantly employed either accuracy or EER to assess model performance, as shown in the Related Works section and smmarized in Table 1. We adopted a more comprehensive approach by using AUC, precision, recall, F1-score, accuracy, and EER to ensure the robustness of the model's performance. These metrics provide a holistic evaluation of the system's effectiveness in keystroke biometric user authentication.

Accuracy, as shown in Eq. (1) provides an overarching view of the system's overall correctness.

$$Accuracy = \frac{TP + TN}{TP + TN + FP + FN}. \tag{1}$$

Equal error rate (EER) is pivotal in measuring the balance between false acceptance rate (FAR) shown in Eq. (2) and false rejection rate (FRR) shown in Eq. (3). A lower EER indicates superior performance as the system maintains a precise equilibrium between authorizing genuine users and rejecting impostors.

$$False\ Acceptance\ Rate = \frac{FP}{FP+TN} \tag{2}$$

$$False\ Rejection\ Rate = \frac{FN}{FP+TN}. \tag{3}$$

Area under the ROC Curve (AUC) quantifies the model's ability to distinguish between genuine and impostor users. A higher AUC value signifies superior discrimination, with values closer to 1 indicating exceptional performance.

These equations express the calculation of false acceptance rate and false rejection rate, which are integral in determining the EER. The EER corresponds to the point on the ROC curve where the false acceptance and false rejection rates are equal. A smaller EER corresponds to higher accuracy in biometric systems.

Precision measures the proportion of true positives out of all positive predictions:

$$Precision = \frac{TP}{TP+FP}. \tag{4}$$

Recall measures the proportion of true positives out of all actual positives:

$$Recall = \frac{TP}{TP+FN}. \tag{5}$$

F1-score is the harmonic mean of precision and recall:

$$F1\text{-}score = \frac{2 \times Precision \times Recall}{Precision+Recall}. \tag{6}$$

The performance of the outlier detection models was evaluated using three key metrics: EER, AUC, accuracy (ACC), precision, recall, and F1-score. EER measures the point at which the FAR equals the FRR, providing a balanced assessment of the system's performance. AUC quantifies the model's ability to distinguish between genuine users and impostors, with higher values indicating better discrimination. ACC represents the overall accuracy of the system.

## Outlier detection model results

In the comprehensive evaluation, a total of 15 different outlier detection models were rigorously tested using the CMY keystroke biometric benchmark dataset. Widely recognized in the field of keystroke dynamics, this dataset provided a robust foundation for assessing the effectiveness of various models. The results of the evaluation are synthesized and presented in Table 2, enabling a quick comparison of the models' performance metrics. These models encompass a range of techniques and algorithms, each with its unique approach to identifying anomalies in keystroke dynamics data.

Table 2 provides a comprehensive comparison of the performance of 15 outlier detection models in the context of keystroke biometric authentication using the CMU dataset. The

**Table 2   Comparison of 15 outlier detection models in keystroke biometric authentication using the CMU dataset.**

| Model name | EER | ACC | AUC |
|---|---|---|---|
| Histogram-base Outlier Detection (HBOS) | 6.42% | 90.55% | 97.68% |
| Isolation Forest | 8.59% | 87.60% | 96.55% |
| Principal Component Analysis (PCA) | 10.45% | 79.39% | 94.74% |
| Gaussian Mixture Model | 11.00% | 81.90% | 94.44% |
| Average KNN | 12.60% | 66.24% | 93.86% |
| Angle-based Outlier Detector (ABOD) | 12.67% | 56.24% | 93.84% |
| K Nearest Neighbors (KNN) | 13.30% | 65.30% | 93.30% |
| Cluster-based Local Outlier Factor (CBLOF) | 14.42% | 64.26% | 91.94% |
| Kernel Density Estimation | 15.62% | 61.36% | 90.11% |
| Feature Bagging | 17.03% | 57.98% | 89.64% |
| INNE | 17.26% | 70.65% | 90.02% |
| Local Outlier Factor (LOF) | 17.60% | 56.93% | 89.09% |
| Locally Selective Combination (LSCP) | 17.86% | 58.60% | 89.03% |
| One-class SVM (OCSVM) | 19.29% | 60.35% | 84.97% |
| LMDD | 44.95% | 54.57% | 60.96% |

**Notes:**
EER, Equal error rate; ACC, Accuracy; AUC, Area under the ROC curve.

results highlight the effectiveness of various models in distinguishing genuine users from impostors based on their keystroke patterns. Notably, HBOS emerges as the top-performing model with the lowest EER of 6.42%, indicating its robust capability to achieve a balance between false positives and false negatives. Following closely are isolation forest and PCA with EERs of 8.59% and 10.45%, respectively, showcasing their competence in the keystroke authentication domain.

Gaussian mixture model and average KNN also exhibit commendable performance with EERs below 12%, suggesting their effectiveness in discerning legitimate users from potential impostors. However, it is worth noting that some models, such as LMDD, show significantly higher EERs, indicating potential challenges in accurately identifying outliers within the keystroke dataset.

Furthermore, to gain deeper insights into the models' performance across various categories and scenarios, receiver operating characteristic (ROC) curves were generated, as visually depicted in Fig. 3. These curves vividly illustrate each model's ability to discriminate between genuine users and potential impostors, offering insights into variations in their effectiveness and characteristics.

The ROC curves in Fig. 3 provide a comprehensive overview of the true positive rates (TPR) and false positive rates (FPR) for each tested outlier detection technique, categorized into linear models, proximity-based models, probabilistic models, and ensembles and combination frameworks. Notably, the ROC curves highlight that the HBOS, isolation forest, and PCA models exhibit the highest TPR and the lowest FPR, indicating their superior effectiveness in distinguishing genuine users from impostors. Conversely, other models display lower TPR and higher FPR, suggesting reduced efficacy in this authentication task. This nuanced analysis using ROC curves enables a more granular understanding of

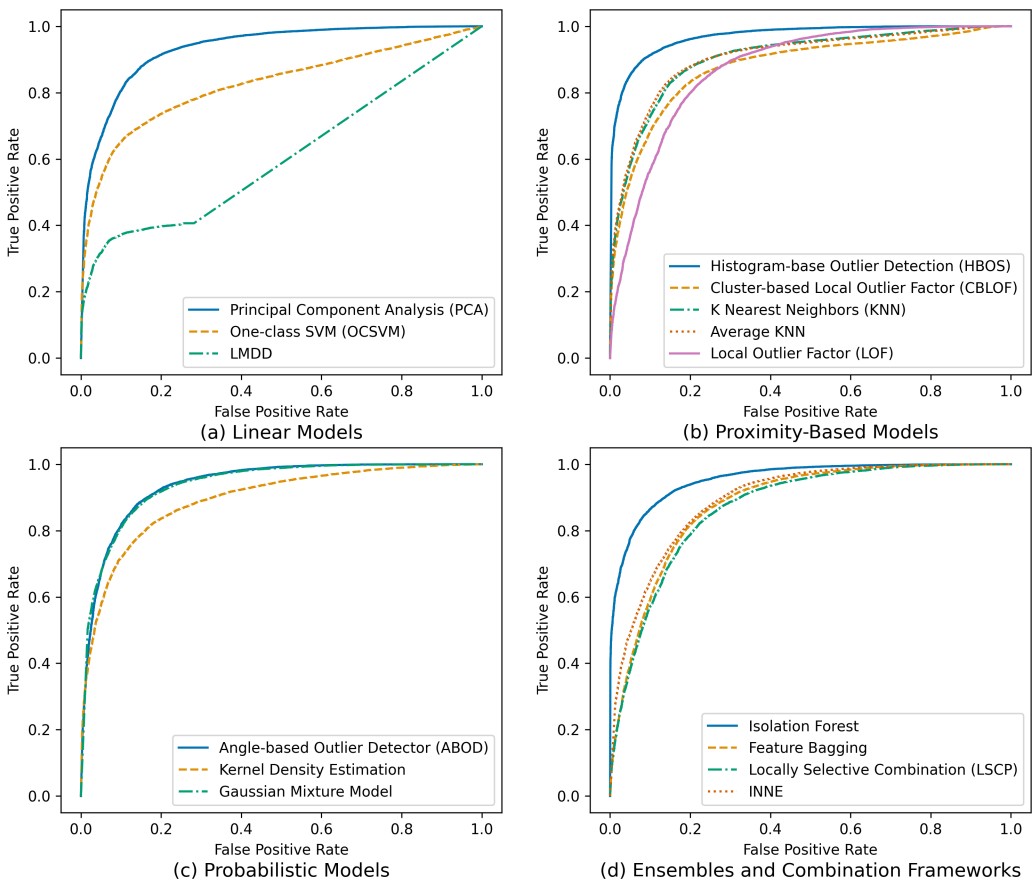

**Figure 3** Receiver operating characteristic (ROC) curves for tested outlier detection techniques.

each model's discriminatory power and guides the selection of the most suitable techniques based on specific security requirements.

## Hyper-parameter tuning results

Hyper-parameter tuning aimed to optimize the performance of various outlier detection models, selected based on their effectiveness and widespread use in outlier detection tasks. These models include HBOS, isolation forest, PCA, LOF, and OCSVM. Each model offers unique strengths: HBOS for efficient high-dimensional outlier detection, Isolation Forest for handling large datasets, PCA for simple dimensionality reduction, LOF for local outlier detection, and OCSVM for detecting outliers in high-dimensional spaces.

For HBOS, parameters such as the number of bins and alpha were tuned. Isolation forest's tuning involved adjusting the number of estimators and maximum samples, among others. PCA's parameters included the number of components and whitening parameter. LOF's tuning focused on the number of neighbors and leaf size, while OCSVM's tuning involved parameters like kernel type, gamma, and nu. This tuning process aimed to enhance the models' accuracy in identifying outliers within keystroke biometric data. The best parameters for the tuned outlier detection models are available in Table 3.

**Table 3  Best parameters for tuned outlier detection models.**

| Model name | Parameters |
|---|---|
| HBOS | [n_bins=15, alpha=0.4, tol=0.3, contamination=0.01] |
| IForest | [n_estimators=400, max_samples=0.5, max_features=0.5, bootstrap=True, contamination=0.01, n_jobs=-1, random_state=42] |
| Local outlier factor | [n_neighbors=15, leaf_size=10, metric=manhattan, $p=1$, contamination=0.01] |
| OCSVM | [kernel=rbf, gamma=scale, nu=0.9, coef0=0.0, contamination=0.01] |
| Principal component analysis | [n_components=5, whiten=True, svd_solver=auto, contamination=0.01, random_state=42] |

**Table 4  Comparison of five tuned outlier detection models in keystroke biometric authentication using the CMU dataset.**

| Model name | EER | ACC | AUC | Precision | Recall | F1-score |
|---|---|---|---|---|---|---|
| HBOS | 5.97% | 89.23% | 97.79% | 75.77% | 98.8% | 85.76% |
| IForest | 7.81% | 89.37% | 97.05% | 92.91% | 94.4% | 93.65% |
| Principal component analysis | 10.43% | 79.50% | 94.69% | 98.84% | 68.0% | 80.57% |
| Local outlier factor | 14.71% | 67.75% | 92.03% | 100.00% | 38.4% | 55.49% |
| OCSVM | 15.52% | 76.30% | 90.40% | 93.89% | 86.0% | 89.77% |

Notes:
EER, Equal error rate; ACC, Accuracy; AUC, Area under the ROC curve.

Additionally, Table 4 presents a comparison of the performance of these models in keystroke biometric authentication using the CMU dataset. The evaluation metrics include EER, ACC, AUC, precision, recall, and F1-score.

In summary, HBOS achieved the lowest EER of 5.97%, indicating its effectiveness in balancing false acceptance and false rejection rates. Isolation forest demonstrated a slightly higher EER of 7.81% but excelled in terms of ACC and AUC, achieving 89.37% and 97.05%, respectively. LOF struggled with a higher EER of 14.71% and relatively lower ACC and AUC. OCSVM and PCA fell in between, with OCSVM showing better performance in terms of ACC and PCA in terms of AUC. Overall, the choice of outlier detection model should consider the trade-offs between different evaluation metrics based on specific application requirements.

Figure 4 presents a comparison of ROC curves for the tuned outlier detection models. Figure 4A displays the collective ROC curves of the tuned outlier detection models, while Fig. 4B showcases individual ROC curves for the 51 users using tuned histogram-based outlier detection. This comprehensive evaluation offers valuable insights into the strengths and limitations of various outlier detection approaches for keystroke biometric authentication.

To highlight the robustness of the histogram-based outlier detection technique, we visualized individual ROC curves for each of the 51 users, depicted in Fig. 4B. This detailed visualization allows for a user-specific assessment, showcasing the technique's consistency across a diverse user base. The impressive average AUC of 97.8% across all

users confirms the effectiveness and practicality of this approach for keystroke biometric user authentication.

This comparison aids security practitioners and researchers in making well-informed decisions by providing a detailed performance overview for each model. Moreover, it delves into the consistency of each model across diverse user behaviors, a crucial aspect for real-world applicability.

The results not only validate the efficacy of unsupervised outlier detection techniques, particularly HBOS, but also address challenges in acquiring representative imposter datasets. These methods offer robust, accurate solutions for real-world applications, underscoring their significance in enhancing security measures amidst the growing reliance on keystroke dynamics for authentication.

## Discussion

This section critically evaluates the study's findings and methodology. Comparing our proposed HBOS with recent keystroke dynamics-based authentication methods, emphasizing its practical advantages. We also highlight limitations, including dataset constraints and the focus on traditional machine learning approaches. Future research directions, particularly exploring deep learning outlier detection methods, are proposed to enhance keystroke biometric authentication systems.

### *Comparison with previous works*

Table 5 provides a comprehensive overview of recent keystroke dynamics-based authentication methods, focusing on their EER using the CMU keystroke dataset. Remarkably, the proposed HBOS not only exhibits competitive performance with an EER of 5.97% and an accuracy of 89.23%, but it also stands out for its distinctive characteristic of not requiring imposter data during training (**x** in the 'Requires Imposter Data' column). This sets HBOS apart from various existing methods, such as histogram gradient boosting (*Ibrahim et al., 2023*), deep secure (*Maheshwary, Ganguly & Pudi, 2017*), MLP (*Andrean, Jayabalan & Thiruchelvam, 2020*), autoencoder model (*Patel et al., 2019*), dependence clustering + KNN (*Ivannikova, David & Hämäläinen, 2017*), and X-means with QT (*Hazan, Margalit & Rokach, 2021*), all of which necessitate imposter data for training (indicated by ✓ in the same column). The results further emphasize the practical applicability and efficiency of HBOS, achieving comparable performance to methods requiring imposter data, while concurrently mitigating the challenges associated with imposter data collection. This underlines the significance of HBOS as a robust and viable solution in keystroke dynamics-based authentication.

In summary, the HBOS not only demonstrates competitive performance but also introduces a significant practical advantage by operating independently of an imposter dataset. This positions HBOS as a promising and efficient solution for real-world security applications, addressing critical challenges associated with imposter data collection. The findings underscore the potential of HBOS to enhance the reliability and applicability of keystroke dynamics authentication in diverse and challenging environments.

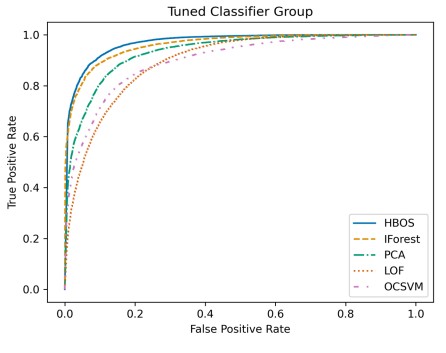

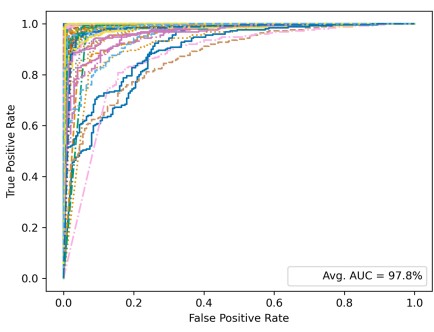

(a) ROC curves of the tuned outlier detection models | (b) ROC curves for 51 users using Tuned HBOS

**Figure 4** **Comparison of ROC curves for tuned outlier detection models.**

**Table 5** **Comparison with recent works using the CMU keystroke dataset sorted by EER score.**

| Author | Approach | Performance | | Requires imposter data |
|---|---|---|---|---|
| | | **EER** | **Accuracy** | |
| *Ibrahim et al. (2023)* | Histogram Gradient Boosting | 1.4% | 97.96% | ✓ |
| *Maheshwary, Ganguly & Pudi (2017)* | Deep Secure | 3% | 93.59% | ✓ |
| *Andrean, Jayabalan & Thiruchelvam (2020)* | MLP | 4.45% | _ | ✓ |
| **Proposed** | Histogram-Based Outlier Score | 5.97% | 89.23% | × |
| *Patel et al. (2019)* | Autoencoder model | 6.51% | _ | ✓ |
| *Ivannikova, David & Hämäläinen (2017)* | Dependence Clustering + KNN | 7.7% | _ | ✓ |
| *Bhatia et al. (2018)* | Generalized Fuzzy Model (GFM) | 7.86% | _ | × |
| *Ali & Tappert (2018)* | POHMM+SVM | 8.6% | _ | × |
| *Alpar (2021)* | Scalogram Barcoding and One-class SVM | 9.88% | _ | × |
| *Hazan, Margalit & Rokach (2021)* | X-means with QT | 11.2% | _ | × |
| *Krishna & Ravi (2019)* | Modified Differential Evolution | 34.8% | _ | × |

### *Limitations*

The first limitation pertains to the dependency of user keystroke biometrics on the keyboard/device used and the user's familiarity with it. In this study, we constrained the environment to a single device and password across all recorded iterations using the CMU dataset. An enhancement in this regard could involve enriching the dataset with multi-device and multi-password cases, either by collecting new data or by amalgamating multiple open-source datasets. This broader dataset would better reflect real-world scenarios, thereby improving the generalizability of the findings.

Another limitation lies in the focus solely on machine learning outlier detection approaches. While these approaches are well-established, recent studies, as evidenced in Table 1, have increasingly delved into deep learning outlier detection methods. Consequently, some of the models utilized in this study may be considered outdated in comparison. To address this limitation, future research should explore deep learning outlier detection approaches and compare their performance with the traditional machine learning

models documented here. Such an investigation would entail evaluating these deep learning models using the same evaluation metrics applied in this study and considering factors such as training and inference time, which are crucial in authentication systems. By embracing the advancements in deep learning techniques, researchers can potentially uncover novel insights and improve the efficacy of keystroke biometric authentication systems.

## CONCLUSION

User authentication stands as a pivotal safeguard against unauthorized access, forming the bedrock of secure systems. Leveraging keystroke biometrics, which capitalizes on the nuances of typing behavior, offers a promising avenue for user identification and authentication.

This article has made significant strides in this domain by investigating the application of unsupervised outlier detection techniques to keystroke biometric authentication, with a specific focus on the histogram-based outlier score (HBOS) algorithm. Through a comprehensive evaluation comparing HBOS to 15 other outlier detection methods, our study has demonstrated its exceptional performance, boasting an equal error rate (EER) of 5.97%, an AUC of 97.79%, and an accuracy (ACC) of 89.23%. Notably, HBOS's ability to operate without an imposter dataset addresses a critical limitation in keystroke dynamics research, enhancing its practical applicability in real-world scenarios.

Looking forward, the research landscape should prioritize addressing the challenge of multi-user authentication, where multiple individuals may share a single account. This necessitates the development of more sophisticated authentication methods capable of reliably distinguishing between authorized and unauthorized users. Additionally, the transition from traditional one-time password checks to a continuous authentication paradigm holds immense promise. By continuously evaluating a user's authenticity throughout their active session, keystroke dynamics can significantly enhance security and user experience. This evolutionary step solidifies the role of keystroke dynamics as a cornerstone of future user authentication systems.

Moreover, future studies should explore the integration of deep learning outlier detection approaches alongside traditional machine learning models. By comparing their performance using established evaluation metrics and considering factors such as training and inference time, researchers can gain insights into the suitability of deep learning techniques for keystroke biometric authentication.

In conclusion, this study not only advances our understanding of keystroke biometric authentication but also paves the way for future research endeavors aimed at fortifying user authentication mechanisms in an increasingly digital landscape.

### Funding

This work was supported by Princess Nourah bint Abdulrahman University Researchers Supporting Project Number (PNURSP2024R409), Princess Nourah bint Abdulrahman

University, Riyadh, Saudi Arabia. The funders had no role in study design, data collection and analysis, decision to publish, or preparation of the manuscript.

### Grant Disclosures

The following grant information was disclosed by the authors:

Princess Nourah bint Abdulrahman University Researchers Supporting Project Number, Princess Nourah bint Abdulrahman University, Riyadh, Saudi Arabia: PNURSP2024R409.

### Competing Interests

The authors declare that they have no conflicts of interest.

### Author Contributions

- Mahmoud G. Ismail conceived and designed the experiments, performed the experiments, analyzed the data, performed the computation work, prepared figures and/or tables, authored or reviewed drafts of the article, and approved the final draft.
- Mohammed A.-M. Salem conceived and designed the experiments, performed the experiments, performed the computation work, prepared figures and/or tables, and approved the final draft.
- Mohamed A. Abd El Ghany conceived and designed the experiments, performed the experiments, performed the computation work, prepared figures and/or tables, and approved the final draft.
- Eman Abdullah Aldakheel performed the experiments, analyzed the data, authored or reviewed drafts of the article, and approved the final draft.
- Safia Abbas performed the experiments, analyzed the data, authored or reviewed drafts of the article, and approved the final draft.

### Data Availability

The CMU keystroke dataset and dataset description are available in the Supplemental File. The dataset is also available at: https://www.cs.cmu.edu/~keystroke.

The code is available at GitHub and Zenodo:

- https://github.com/Mahmoud-GIsmail/Outlier-Detection-for-Keystroke-Dynamics
- Mahmoud Gamal. (2024). Mahmoud-GIsmail/Outlier-Detection-for-Keystroke-Dynamics: Published version (Version1.0.0). Zenodo. https://doi.org/10.5281/zenodo.11144535.

A GitHub Zip folder contains a Jupyter notebook file with the implemented code for the study. This folder includes all files and functions used for data processing, model training, and evaluation, allowing for full reproducibility of the results presented in the article. The Jupyter notebook provides detailed documentation and instructions for running the code, ensuring that other researchers can easily follow and replicate the procedures described in our study.

### Supplemental Information

Supplemental information for this article can be found online at http://dx.doi.org/10.7717/peerj-cs.2086#supplemental-information.

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
