# Peer review of "Outlier detection for keystroke biometric user authentication"

_PeerJ Computer Science, doi:10.7717/peerj-cs.2086_

## Round 0.1 · original submission · Minor Revisions

The paper lacks a clear discussion on false positives, such as the impact of keyboard types and typing speed, which could affect system accuracy. Reviewers recommend refining the experimental design to better evaluate the proposed method's effectiveness, including addressing the challenge of obtaining authentic imposter data. Furthermore, the paper lacks detailed model configurations and comparisons with newer outlier detection techniques, potentially limiting the validity of its findings. Suggestions to address these concerns include providing clearer discussions on false positives, refining experimental design and evaluation metrics, detailing model configurations, and incorporating relevant citations while focusing on major contributions. Addressing these concerns will enhance the paper's quality, validity, and impact in the field of keystroke biometric authentication.

**Language Note:** PeerJ staff have identified that the English language needs to be improved. When you prepare your next revision, please either (i) have a colleague who is proficient in English and familiar with the subject matter review your manuscript, or (ii) contact a professional editing service to review your manuscript. PeerJ can provide language editing services - you can contact us at [email protected] for pricing (be sure to provide your manuscript number and title). – PeerJ Staff

Reviewer 1 ·

Basic reporting

This paper presents a new approach to keystroke dynamics-based user authentication, emphasizing the use of unsupervised outlier detection techniques, particularly the Histogram-Based Outlier Score (HBOS), to overcome the challenge of relying on imposter datasets. The conventional method of utilizing imposter data for training authentication systems faces practical difficulties, notably in collecting authentic imposter keystrokes, which this research effectively addresses. The study's motivation stems from the critical demand for enhanced user authentication methods that bypass the limitations of traditional keystroke dynamics techniques, which heavily depend on imposter datasets. Key contributions of this paper include the introduction of an alternative approach in keystroke authentication through unsupervised outlier detection, notably HBOS, compared against 15 other outlier detection methods.

Overall, the paper is well-organized.

Experimental design

The authors may consider adding some experiments, some facts or other evidence that can show "obtaining authentic imposter data is a daunting task, and prevailing methodologies, often relying on simulated fraudulent keystrokes, fall short of accurately representing genuine imposter behavior".

Validity of the findings

no comment

Additional comments

In the introduction section, add some citations while mentioning that related works cannot solve the target problem. For contributions of the work, the authors may consider only listing three major contributions in summary.

Cite this review as

·

Basic reporting

This paper proposes the use of outlier detection for keystroke biometric user authentication, representing a novel combination of machine learning and authentication techniques. The paper is overall well-written, effectively clarifying the motivation and previous works before introducing their new solution, making it easy to follow.

One significant concern I have is regarding the potential false positives associated with using keystroke biometrics for user authentication. The paper does not seem to provide a clear discussion addressing this crucial issue. For instance, the type of keyboard used can play a vital role in introducing false positives. If a user switches from one keyboard to another, or moves from a PC to a laptop, the model may fail to recognize them as the same person due to variations in keystroke distance across different input devices. Another factor to consider is typing speed; typing rapidly often results in smaller time differences in the Up-Down (UD) and Down-Down (DD) intervals, potentially increasing the false positive rate. Unfortunately, the paper fails to elaborate in detail on these confounding factors that could impact the system's accuracy.

Experimental design

While the authors mentioned an office scenario to motivate their work, it would be beneficial to provide a few more sentences constraining other potential real-world factors that could impact the false positive rate. For example, a scenario where all users are using the same type of keyboard could help mitigate the confounding factor of different keyboard types and provide a more controlled environment to evaluate the proposed method's effectiveness.

Validity of the findings

The experiment is overall pleasant, and the outlier detection model presents good results compared to other methods. However, this paper's current version lacks models' detailed configuration. For example, the hyperplane parameters will significantly affect the performance of the One-Class Support Vector Machine (OCSVM), but this paper fails to present the values of these parameters that it's using. It is unclear if the authors conducted an ablation study to determine the optimal hyperparameter settings. Additionally, some of the compared models may seem outdated. It would be beneficial if the authors plan to compare their method with newer one-class classifiers and outlier detection techniques, which could provide a more comprehensive evaluation of their proposed approach's performance.

Additional comments

Overall, the paper is above the bar, and I recommend its acceptance contingent upon the authors addressing the aforementioned concerns regarding false positives, hyperparameter tuning, and real-world scenario considerations.

Cite this review as

Reviewer 3 ·

Basic reporting

The paper provides an investigaition to different outliers method when applied to keystroke authentication. Specially, the work evaluates HBOS and 15 other methods. It provides backgrounds to these methods and shows the evaluation results. Figures are well-suited and revelant to the article. But there are some minor issues about literature references: line 303 states "While prior studies predominantly employed either accuracy or Equal Error Rate", but there is no references.

Experimental design

The methods are evaluated on Carnegie Mellon University (CMU) Keystroke Biometric Benchmark Dataset. But there are some minor issues about the evaluation metrics: precision, recall, F1-score are also important metrics when evaluating machine learning methods. It is better to provide such metrics to make the evaluation complete.

Validity of the findings

No comment

Cite this review as

---

## Round 0.2 · accepted · Accept

The revision addressed all the comments, and thanks for the hard work.

·

Basic reporting

no comment

Experimental design

no comment

Validity of the findings

no comment

Additional comments

Thank you for making efforts to address my concerns, I'm overall optimistic about this paper.

Cite this review as